# TEMPORAL GAUSSIAN MIXTURE LAYER FOR VIDEOS

## ABSTRACT

We introduce a new convolutional layer named the *Temporal Gaussian Mixture* (TGM) layer and present how it can be used to efficiently capture longer-term temporal information in continuous activity videos. The TGM layer is a temporal convolutional layer governed by a much smaller set of parameters (e.g., location/variance of Gaussians) that are fully differentiable. We present our fully convolutional video models with multiple TGM layers for activity detection. The experiments on multiple datasets including Charades and MultiTHUMOS confirm the effectiveness of TGM layers, outperforming the state-of-the-arts.

## 1 INTRODUCTION

Activity videos are spatio-temporal data: they are image frames with a specific width/height (XY) concatenated along time axis (T). Recognition from such videos requires capturing both spatial and temporal information in the videos, desirably using learned convolutional kernels. Temporal convolution is particularly beneficial in activity 'detection' tasks, which require making activity decisions at every frame given a *continuous* video (Sigurdsson et al., 2016b; Yeung et al., 2015). Previous methods investigated using 3-D XYT convolutional filters (Tran et al., 2014; Carreira & Zisserman, 2017) as well as the models with 2-D XY conv. layers followed by 1-D temporal conv. (Tran et al., 2018), pooling or attention layers (Piergiovanni et al., 2017).

Understanding complex multi-activity videos requires capturing information in long-term time intervals. Different frames contain different information, and the model needs to learn to take advantage of as many frames as possible, while abstracting them efficiently. Previous attempts of simply pooling representations over time or learning temporal conv. filters with a small number of frames (e.g., 16 or 64) was thus often insufficient to fully consider rich long-term temporal context. Simultaneously, bruteforcely increasing the temporal filter length (to look at more frames) results more learnable parameters, requiring more training data, which can be expensive when activities are rare.

In this paper, we introduce a new convolutional layer named the *Temporal Gaussian Mixture* (TGM) layer, and present how it can be used to efficiently capture longer-term temporal information in activity videos. Our temporal Gaussian mixture layer is a temporal convolutional layer, whose filters/kernels are controlled by a set of (temporal) Gaussian distribution parameters. Each of our temporal Gaussian distributions specify (temporally) 'where' the model should look, and our Gaussian mixture layer combines them as multiple convolutional filters to be applied on top of temporally-continuous representations. This layer allows the video representation at each time step to be constructed while focusing on different neighboring temporal regions, instead of only focusing on its local segment. It is a convolutional layer governed by a much smaller set of parameters (i.e., locations/variances of the Gaussians as well as their mixture weights) that are fully differentiable.

The motivation behind our temporal Gaussian mixture layer is to learn the temporal structure of an activity as a composition of temporal Gaussian regions/attentions. Such structure allows the model to obtain a compact spatio-temporal representation abstracting each (long-term) time interval, using multiple temporal conv. layers with far fewer parameters. It is also related to the previous temporal attention works (Piergiovanni et al., 2017), but our model is designed to be fully convolutional to handle continuous data and it learns more compositional structures with multiple layers.

We present video-CNN models using our TGM layers for activity detection in continuous videos. Our model stacks TGM layers on top of several state-of-the-art CNNs such as I3D (Carreira & Zisserman, 2017). This enables our model to capture longer-term temporal information than what we use as base CNNs, compositionally modeling temporal structure with multiple TGM layers. Our

model was evaluated on multiple public datasets including MultiTHUMOS and Charades, and was able to outperform the best previous activity detection CNNs by a meaningful margin.

## 2 RELATED WORKS

Learning video representations for human activity recognition has been successful. CNN methods allow end-to-end learning of video features and representations optimized for the training data, performing superior to traditional works (Aggarwal & Ryoo, 2011) for video understanding.

Two-stream CNN models take a single RGB frame and a small number of optical flow frames as inputs to capture both motion and appearance information in videos (Simonyan & Zisserman, 2014; Feichtenhofer et al., 2016). Models learning 3-D spatio-temporal (XYT) convolutional filters were designed and applied to many activity recognition tasks as well (Tran et al., 2014; Carreira & Zisserman, 2017; Tran et al., 2017; Hara et al., 2017). Large scale datasets for activity detection, such as THUMOS (Jiang et al., 2014), ActivityNet (Heilbron et al., 2015), Kinetics (Kay et al., 2017), and Charades (Sigurdsson et al., 2016b) provided these approach the necessary training data to learn the models. Such 3-D XYT CNNs were also used to capture spatio-temporal information for activity detection (Xu et al., 2017; Shou et al., 2016; 2017; Zhao et al., 2017). However, all these CNNs were limited to the consideration of a fixed local video segment (e.g., 16 frames in (Tran et al., 2014) and 64-99 frames in (Carreira & Zisserman, 2017)) when making activity decisions.

Some works studied combining representations over longer-term temporal intervals (Karpathy et al., 2014; Ng et al., 2015; Varol et al., 2017), but it was generally done with a temporal pooling of local representations or (spatio-)temporal convolutions with a bit larger fixed intervals. Recurrent neural networks (RNNs) have also been used to model activity transitions between frames (Yeung et al., 2015; 2016; Escorcia et al., 2016), but they were strictly sequential and had limitations in maintaining temporal information over a longer temporal duration, particularly for videos with multiple complex activities. Recently, CNN models using temporal attention for activity videos (Piergiovanni et al., 2017; Piergiovanni & Ryoo, 2018b) were studied as well. However, a fully convolutional model to analyze continuous videos while efficiently representing information in long term intervals has been lacking.

Our layer is different from the previous standard (spatio-)temporal convolutional layers in that it relies on significantly fewer parameters by forcing filter shapes to be Gaussian compositions. Our temporal layer is also different from previous Gaussian Mixture Model layers (Variani et al., 2015) in that our layer is convolutional while they are not.

## 3 APPROACH

In this section, we introduce a new convolutional layer named the *Temporal Gaussian Mixture* (TGM) layer, and present how it can be used for activity recognition. Our Temporal Gaussian Mixture layer is a temporal convolutional layer to be applied on top of a sequence of representations (usually from frame-level or segment-level CNNs), whose filters/kernels are controlled by a set of (temporal) Gaussian distribution parameters. The motivation is to make each temporal Gaussian distribution specify (temporally) 'where to look' with respect to the activity center, and represent the activity as a collection/mixture of such temporal Gaussians convolved with video features. Our layer is fully differentiable and trainable using standard backpropagation.

Our TGM layer can be interpreted as a a form of 1-D convolution where the filters are determined by a mixture of Gaussians. However, our TGM layer differs from the standard temporal convolutional layers of learning 1-D (time) or 2-D (channel-by-time) filters in the following aspects:

1. Our temporal Gaussian mixture layer handles multiple 3-D tensors internally to preserve channels from the frame-level CNN by adding a new temporal channel axis. Its input is 3-D (channel-by-channel-by-time), where one channel dimension is inherited from the frame-level CNN and this dimension size remains unchanged.

2. Instead of learning temporal convolution filters of any arbitrary values, our filter is forced to have the form of a temporal Gaussian mixture shared across all frame-level channels. This allows the layer to rely on significantly fewer number of (fully differentiable) parameters, while capturing the concept of temporal structure/attention.

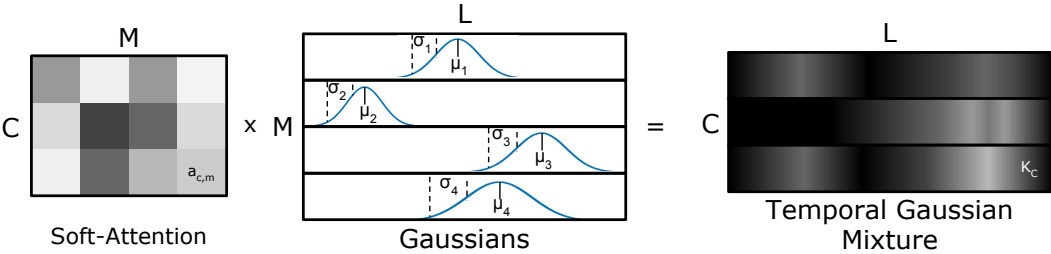

Figure 1: Example illustrating how our Temporal Gaussian Mixture layer is computed. Multiple ($M$) temporal Gaussian distributions are learned, and they are combined with the learned soft attention weights to form the $C$ temporal convolution filters. $L$ is the temporal length of the filter.

## 3.1 TEMPORAL GAUSSIAN MIXTURE LAYER

Our temporal Gaussian mixture layer takes a 3-D input with the dimensionality of $C_{in} \times D \times T$, where $C_{in}$ is the number of input channels, $D$ is the dimensionality of the representations from frame-level (or segment-level) CNNs, and $T$ is the time. Given such input, the TGM layer convolves it with $C_{out}$ number of $1 \times L$ filters/kernels, generating a $C_{out} \times D \times T$-dim representation as an output. $L$ is the temporal length of the temporal Gaussian mixture filter. $D$ is usually 1K or 4K and $T$ is the number of time steps (frames) in each video (i.e., it varies per video). $C_{out}$ is the number of different mixtures, corresponding to the number of output channels in standard convolution.

Our layer is composed of a set of $M$ Gaussians. Each Gaussian has 2 parameters: a center $\hat{\mu}$ and a width $\hat{\sigma}$. Each layer has additional hyper-parameters: $L$, the temporal duration and $M$, the number of Gaussians to learn. We force the learned center to be between $-\frac{L}{2}$ and $\frac{L}{2}$ and $\sigma$ to be positive:

$$\mu = (L-1) \cdot \frac{\tanh\left(\hat{\mu}+1\right)}{2}, \;\; \sigma^2 = \exp\left(\hat{\sigma}\right). \tag{1}$$

We use the above $\mu$ and $\sigma$ to construct the temporal Gaussian kernels. This acts as a strong sparsity constraint on the convolutional kernel as well as a drastic reduction of the number of learnable parameters. We construct a temporal Gaussian mixture convolutional kernel as:

$$\hat{K}_{m,l} = \frac{1}{Z} \exp -\frac{(l-\mu_m)^2}{2\sigma_m^2} \tag{2}$$

where $Z$ is a normalization constant such that $\sum_l^L \hat{K}_{m,l} = 1$, resulting in $\hat{K}$ being an $M \times L$ matrix.

Instead of making the model learn a separate set of Gaussian distributions per activity class, we take the approach of maintaining multiple Gaussian distributions shared across classes and obtain a Gaussian 'mixture' filter by learning soft-attention weights. We learn a set of soft-attention weights per output channel $i$, $\omega \in \mathcal{R}^{C_{out} \times M}$. We create the soft-attention weights by applying the softmax function over the $M$ Gaussians, enforcing each input channel weights sum to 1.

$$a_{i,m} = \frac{\exp \omega_{i,m}}{\sum_j \exp \omega_{i,j}} \tag{3}$$

Based on temporal Gaussian distributions $\hat{K}_i$ and attention weights $a_{i,m}$, the temporal convolution filters our TGM layer is computed as:

$$K_i = \sum_m a_{i,m} \hat{K}_i. \tag{4}$$

This provides us convolutional filters having the form of a mixture of temporal Gaussians, controlled based on $2 \cdot M + C_{in} \cdot C_{out} \cdot M$ parameters (instead of learning $D^2 \cdot L$ parameters without any constraint, as in standard temporal convolution where $C << D$). An overview of this process is shown in Fig. 1.

### 3.1.1 SINGLE TGM LAYER - DIRECT PER-CLASS ACTIVITY MODELING

The representation we obtain by applying our base CNNs to each frame (or local segment) has the dimensionality of $D$, and stacking them along time axis provides us the representation with

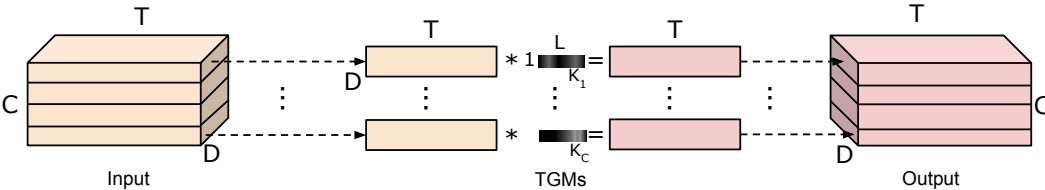

Figure 2: Illustration of a TGM layer with grouped convolution. This layer learns a set of $C$ Gaussian mixtures that are convolved with the input channels.

$1 \times D \times T$-dim. That is, in the case of using only one TGM layer to capture activity representations, our $C_{in}$ is fixed to 1 and $C_{out}$ is fixed to be the number of activity classes. This is the simplest case of our model, attaching one TGM layer on top of the $1 \times D \times T$ representation.

Our convolutional kernel, $K$, has a learned Gaussian mixture for each activity class. Let the video features $v$ be a $D \times T$ matrix. Each $K_i$ is a 2-D convolutional filter with a size of $1 \times L$, and convolving this with $v$ provides us a representation $S$ with $C_{out}$ number of $D \times T$ responses since $C_{in}$ is 1 in this case. This per-class representation can then be used as input to a fully-connected layer for activity classification. For $i \in \{1, 2, \ldots, C_{out}\}$:

$$s_i = v * K_i, \ \ S = [s_1, s_2, \ldots, s_{C_{out}}] \tag{5}$$

Fig. 7 in the appendix visually illustrates how each TGM filter is convolved with the input (Fig. 7d), compared to the standard 1-D convolution (Fig. 7a) or other forms of the temporal layers (Fig. 7b-c).

### 3.1.2  MULTIPLE TGM LAYERS - GROUPED CONVOLUTION

We generalize the above formulation to allow the TGM layers to be sequentially applied. The idea is to enable our model to capture more complex, nonlinear temporal structure by having multiple levels of temporal layers. In this case, the input for each layer is $C_{in} \times D \times T$ dimensional (instead of $1 \times D \times T$), where the input channels are the number of output channels from the previous layer. Our kernels at each layer, $K_i$, are parameterized and learned as before.

By using grouped convolution with the number of groups set to $C_{in}$, we can efficiently separate the input into per-channel values and convolve each of them with the designated $K_i$ kernel, as shown in Fig. 2. That is, we learn a filter $K_i$ per channel by setting $C_{in} = C_{out}$. For $i \in [1, C_{out}]$,

$$s_i = f_i * K_i, \ \ S = [s_1, s_2, \ldots s_{C_{out}}] \tag{6}$$

Here, $f$ is a $C_{in} \times D \times T$ tensor, where $D$ is the dimensionality of the feature and $T$ is the number of frames. The result of the per-channel convolution, $s_i$, is a $D \times T$ representation. We concatenate these representations along the channel axis, resulting in $S$, a $C_{out} \times D \times T$ representation. As this convolution results in the same output shape, we can stack these layers. Each layer is able to capture increasing temporal resolution, allowing the model to capture levels of abstractions.

### 3.1.3  MULTIPLE TGM LAYERS - CHANNEL COMBINATION

In the above subsection, we introduced an approach of stacking multiple TGM layers to model a hierarchical composition of temporal representations. However, in the grouped convolution case, each output channel of the layer is solely dependent on its corresponding input channel. That is, each kernel only considers information from a single output channel of the previous layer.

Therefore, we further generalize our TGM layer so that the layer combines representations from multiple input channels for each output channel while using the learned temporal kernels. We learn a set of convolutional kernels $K \in \mathcal{R}^{C_{out} \times C_{in} \times L}$ (i.e., we learn $C_{out} \cdot C_{in}$ Gaussian mixtures). Given $f$ which is the $C_{in} \times D \times T$ representation, for each output channel $i \in [1, C_{out}]$ and each input channel $j \in [1, C_{in}]$ pair, we convolve the associated filters with the input.

$$G_{i,j} = (f_j * K_{i,j}) \tag{7}$$

where each $G_{i,j}$ is a $D \times T$-dim representation.

We then learn a 1x1 convolution followed by a ReLU activation function for each $i \in [1, C_{out}]$, which we call $w_i$, that maps from $C_{in}$ channels to 1 channel. The 1x1 convolution learns to combine

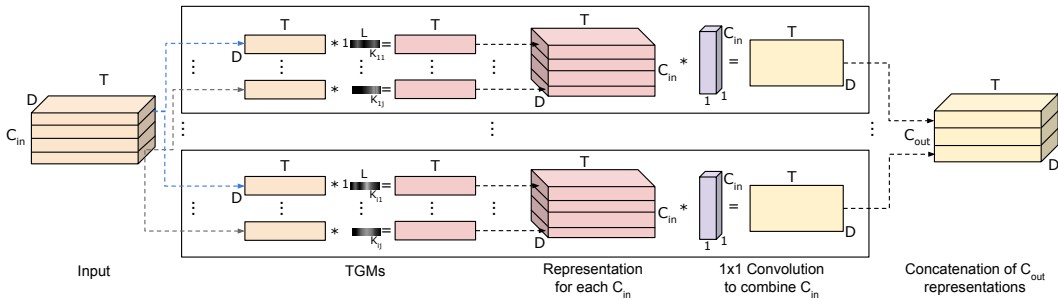

Figure 3: Illustration of a TGM layer with channel combination. The kernels are applied to each input channel, $C_{in}$, and a 1x1 convolution is applied to combine the $C_{in}$ input channels for each output channel, $C_{out}$.

the channels from the previous layer. By design, the TGM kernel is positive and sums to 1. Adding the unconstrained 1x1 convolution adds non-linearity (using the ReLU activation function) to our layer and only adds $C_{out} \cdot C_{in}$ parameters.

$$s_i = G_i * w_i = (f_j * K_{i,j}) * w_i, \quad S = [s_1, s_2 \ldots, s_{C_{out}}] \tag{8}$$

We then stack the $s_i$ representations along the channel axis to produce $S$, the $C_{out} \times D \times T$-dim representation. This process is illustrated in Fig. 3. This method generalizes our approach to allow the layer to take input of $C_{in} \times D \times T$ and produce output of $C_{out} \times D \times T$. These layers can easily be stacked to learn a hierarchical representation.

## 3.2 VIDEO CNN MODELS WITH TGM LAYERS

Our goal is to do activity detection which we define as making a per-frame (or per-segment) classification. Given a video, at each time step $t$, we want to make the model decide which activity the frame corresponds to (including no-activity). As a baseline, we train a fully-connected layer that classifies each per-frame $D$-dimensional vector, $v_t$. As multiple activities can occur at the same time, or no activities at all, we treat this as a mutli-label classification task. We minimize binary cross entropy:

$$L(v) = \sum_{t,c} z_{t,c} \log(p(c|v_t)) + (1 - z_{t,c}) \log(1 - p(c|v_t)) \tag{9}$$

where $z_{t,c}$ is the ground truth label, 1 if activity $c$ is occurring at time $t$ and $p(c|v_t)$ is the output of our model for class $c$ at time $t$. Fig. 4 shows an example CNN.

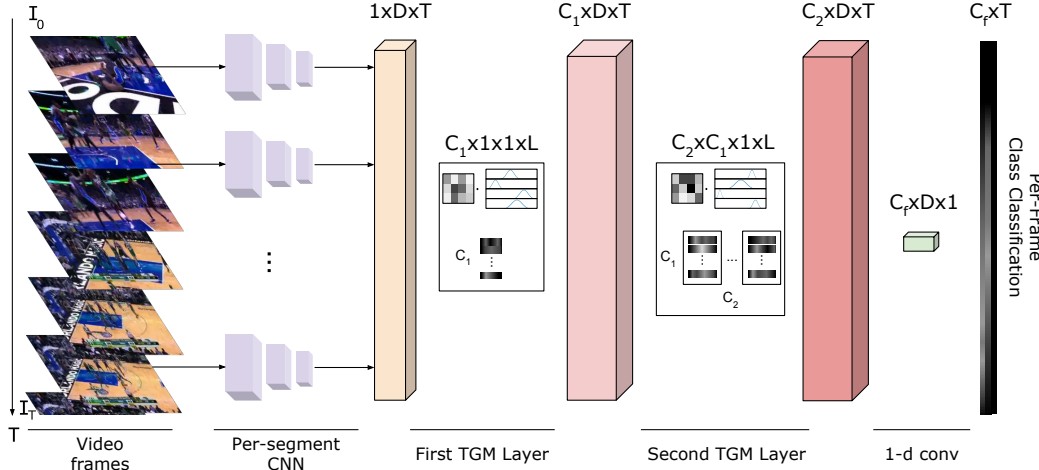

Figure 4: An overview of an example video CNN model with two TGM layers. It is able to handle videos with any length, because of its fully convolutional design.

# 4 EXPERIMENTS

## 4.1 IMPLEMENTATION AND BASELINES

**Implementation**   We used I3D (Carreira & Zisserman, 2017) and the two-stream version of InceptionV3 (Szegedy et al., 2016) pretrained on Imagenet and Kinetics as our base per-frame CNNs. Our default $L$ setting used for the TGM layers as well as the other baselines was as follows: when using I3D segment features (collected at 3fps), the 1 layer models used $L = 15$ and the 3 layer models used $L = 5$. When using InceptionV3 frame feature (collected at 8fps), the 1 layer models used $L = 30$ and the 3 layer models used $L = 10$. These layers were attached on top of the base CNN, as described in Subsection 3.2. Please check the appendix for implementation and training details and results on other datasets.

**Baselines**   In order to confirm the advantages of our TGM layers, particularly against previous temporal models, we implemented several baselines. The first is (i) a standard per-frame classifier in which the prediction at each time-step only depends on a single feature vector with no contextual temporal information. We also used (ii) LSTMs on top of per-frame representations, which were popularly used to capture temporal information (Donahue et al., 2015). We train a bi-directional LSTM with 512 hidden units to make per-frame predictions. We also tried (iii) the fixed pyramid temporal max-pooling of level 3 (Ryoo et al., 2015). Finally, we compare our model against (iv) the model with standard temporal convolutional layers (i.e., 1-D convolution with a $D \times L$ kernel) on top of per-frame representations. This is similar to the temporal conv. used in (Tran et al., 2018). Temporal lengths (i.e., $L$) of the 1-D conv. filters and the pooling windows were set to be identical to the TGM filters. That is, they capture the same temporal duration as TGMs. In all our experiments, we follow the standard evaluation setting of computing per-frame mean average precision (mAP) and report those values. We also compare to different versions of the TGM layer, (v) with a learned mixture of random temporal filters and (vi) with a learned mixture of fixed Gaussians.

In addition, we also tried the approach of combining our TGM layers with the recent super-event representations (Piergiovanni & Ryoo, 2018b). We concatenated the learned super-event representation with our representations from TGM layers.

## 4.2 MULTITHUMOS

**Dataset**   MultiTHUMOS (Yeung et al., 2015) is an extended version of the THUMOS (Jiang et al., 2014) dataset that densely annotates the continuous videos. The dataset consists of 65 different classes, compared to 20 in THUMOS, and contains on average 10.5 activities per video and 1.5 labels per frame and up to 25 activity instances in each video. This is in contrast to many other activity detection dataset such as ActivityNet (Heilbron et al., 2015), which only has on average ∼1 activity per video. MultiTHUMOS consists of YouTube videos of various sport activities such as basketball games, volleyball games, weight lifting, and track and field.

We followed the standard MultiTHUMOS evaluation setting of measuring mAP based on per-frame annotations. There are 1010 validation videos and 1574 test videos. We used these continuous validation videos for the training of our models. We did not need to take advantage of the separate training set with segmented videos; even without them, we outperformed the state-of-the-arts.

**Results**   We compared baselines as well as multiple different versions of our architectures, shown in Table 1. The model with our TGM layers consistently outperformed baseline I3D (or InceptionV3) while using the same per-segment representations. Learning 3 TGM layers further improved the performances. On the other hand, we found that stacking multiple standard temporal convolutional layers does not improve performance, often performing worse than the baseline. While a single standard temporal conv. layer improves over the baseline, having multiple of them significantly increases the number of parameters to learn (Table 2) and we suspect that this was causing the overfitting with the limited amount of samples in the dataset. In Table 3, we compare the results of using a LSTM or temporal conv. with a similar number of parameters. This was done by making their temporal conv. filters to share values across multiple channels. These models result in nearly random performance, as they were not designed to cope with a small number of parameters. We also show results with a mixture of random (fixed) temporal filters and with a mixture of fixed Gaussians. These results confirm that (i) modeling the temporal structure as a learned Gaussian mixture is beneficial and that (ii) further learning the Gaussian distribution parameters is important.

Table 1: Comparison of various architectures on MultiTHUMOS using both I3D per-segment and InceptionV3 per-frame features. We found that TGM layers with 1x1 convolution channel combination performed the best. Results are in mAP %. Note that we use the same filter length for "Temporal Conv" and "TGM" models, as described in Section 4.1.

| | I3D | | | InceptionV3 | | |
| --- | --- | --- | --- | --- | --- | --- |
| | Spatial | Temporal | Two-Stream | Spatial | Temporal | Two-Stream |
| Baseline | 22.3 | 25.0 | 29.7 | 13.6 | 14.1 | 15.2 |
| Temporal Conv | 32.5 | 35.5 | 38.4 | 15.2 | 15.5 | 15.8 |
| 3 Temporal Conv | 20.4 | 23.4 | 24.4 | 5.3 | 6.1 | 6.5 |
| TGM layers with grouped convolution | | | | | | |
| 1 TGM | 35.1 | 37.8 | 40.5 | 16.3 | 17.5 | 18.0 |
| 3 TGM | 36.4 | 42.3 | 43.5 | 17.5 | 18.3 | 19.2 |
| TGM layers with channel combination | | | | | | |
| 1 TGM (soft) | 35.2 | 37.9 | 40.2 | 17.2 | 17.6 | 18.4 |
| 1 TGM (1x1) | 36.1 | 38.2 | 40.8 | 17.2 | 17.7 | 18.4 |
| 3 TGM (soft) | 36.2 | 40.1 | 42.3 | 17.5 | 19.1 | 21.2 |
| 3 TGM (1x1) | 37.2 | 42.1 | **44.3** | 17.9 | 19.3 | 22.2 |

Table 2: Additional number of parameters for models when added to the base architecture (e.g., I3D or Inception V3).

| Model | # of parameters |
| --- | --- |
| LSTM | 10.5M |
| 1 Temporal Conv | 10.5M |
| 3 Temporal Conv | 31.5M |
| 1 TGM Layer | 10K |
| 3 TGM Layers | 100K |

Table 3: Comparison of previous methods with comparable number of parameters and random forms of our TGM layer.

| Model | mAP |
| --- | --- |
| LSTM with 100k parameters | 6.5 |
| Temporal Conv. with 100k parameters | 7.3 |
| TGM with random temporal filters | 34.5 |
| TGM with fixed Gaussians | 38.5 |
| Full TGM | **44.3** |

Learning multiple TGM layers with channel combination outperforms the grouped convolution version of TGM and all the baselines. We also experimented with a version using soft-attention weights to combine the TGM layer channels, in addition to our method (Fig. 3) of using 1x1 convolution followed by a ReLU (to gain non-linearity). We found that the 1x1 convolution performed better. We tested various number of Gaussian mixtures (i.e., output channels) and found that using 80 for the first and second layer and using 65 (i.e., number of classes) for the final layer performs best.

Table 4 compares our model using TGM layers with multiple previous state-of-the-art approaches and baselines such as LSTM. Our approach meaningfully outperforms all previous approaches. Importantly, we are comparing our approach with different methods of capturing temporal information such as LSTMs and fixed temporal pyramid pooling while making them use the exactly same per-frame representations. We found that while all these methods capture some temporal information, the TGM layers provide the best performance. Further, combining the super-event representation (Piergiovanni & Ryoo, 2018b) with our TGM feature also benefited detection, confirming that our TGMs and super-events capture different aspects of the activity videos. In Fig. 5, we show an example of the various models predictions on a basketball video. We outperform the previous state-of-the-art performance (mAP) by 10% (36.4 vs. 46.4).

### 4.3 CHARADES

**Dataset** Charades (Sigurdsson et al., 2016b) is a large scale dataset with 9848 videos across 157 activity classes. These videos were recorded in home environments of the participants based on provided scripts. Each video contains on an average of 6.8 activity instances, and there are often complex activities co-occurring. The activities were mainly performed at home. For example, some activity classes are 'preparing a meal', 'eating', 'sitting', 'cleaning', etc.

In our experiments, we follow the original Charades detection setting (i.e., Charades_v1_localize evaluation), which is the setting used in many previous approaches (Sigurdsson et al., 2016a; Xu

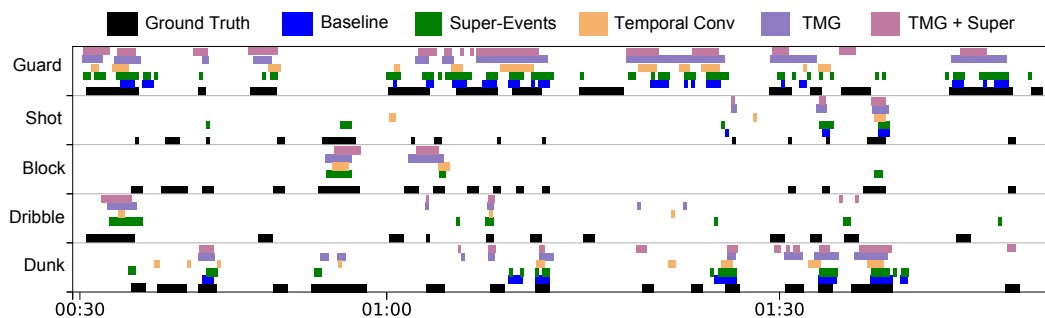

Figure 5: Illustration of the temporal regions classified as various basketball activities from a basketball game video in MultiTHUMOS. Our TGM layers greatly improve performance.

Table 4: Performances of the state-of-the-art methods and our approach on MultiTHUMOS. Our approach meaningfully outperforms all previous results.

|                                                              | mAP      |
| ------------------------------------------------------------ | -------- |
| Two-stream (Yeung et al., 2015)                              | 27.6     |
| Two-stream + LSTM (Yeung et al., 2015)                       | 28.1     |
| Multi-LSTM (Yeung et al., 2015)                              | 29.6     |
| Predictive-corrective (Dave et al., 2017)                    | 29.7     |
| I3D baseline                                                 | 29.7     |
| I3D + LSTM                                                    | 29.9     |
| I3D + temporal pyramid                                       | 31.2     |
| I3D + super-events (Piergiovanni & Ryoo, 2018b)              | 36.4     |
| I3D + our TGMs                                               | 44.3     |
| I3D + super-events (Piergiovanni & Ryoo, 2018b) + our TGMs   | **46.4** |

et al., 2017; Piergiovanni & Ryoo, 2018b). This is the original setting more challenging than the Charades Challenge 2017 setting (whose evaluation server was no longer approving new account access), in the aspect that it uses less amount of training videos.

**Results**  We compare our results with the state-of-the-arts in Table 5. To our knowledge, our method is obtaining the best known performance in the original localization setting of the Charades dataset. Notably, it is performing better than I3D that obtained the best competition performance, while using the same feature. Our method also outperforms standard temporal convolution, LSTMs, and fixed pyramid pooling, as well as the use of latent super-events. When setting $L = 30$ and using 3 TGM layers, our model is able to capture around 800 frames (about $\pm 15$ seconds from each frame) of temporal information, significantly more than previous works (e.g., I3D only captures $\pm 2$ seconds).

## 5    CONCLUSIONS

We newly introduced the Temporal Gaussian Mixture (TGM) layer and demonstrated its effectiveness for multi-activity detection in continuous videos. Our layer is fully differentiable and trainable using standard backpropagation, designed to learn temporal structure. We were able to confirm that our layer performs superior to state-of-the-art methods on activity detection datasets including MultiTHUMOS and Charades, obtaining the best known performance. We also tested our approach with two more public video datasets, MLB-YouTube (Piergiovanni & Ryoo, 2018a) and AVA (Gu et al., 2017), and confirmed its advantage over the previous works in Appendix.

## REFERENCES

J. K. Aggarwal and M. S. Ryoo. Human activity analysis: A review. *ACM Computing Surveys*, 43: 16:1–16:43, April 2011.

Table 5: Per-frame mAP on Charades, evaluated with the 'Charades_v1_localize' setting. I3D models are two-stream, using both RGB and optical flow inputs.

|  | mAP |
|---|---|
| Predictive-corrective (Dave et al., 2017) | 8.9 |
| Two-stream (Sigurdsson et al., 2016a) | 8.94 |
| Two-stream+LSTM (Sigurdsson et al., 2016a) | 9.6 |
| R-C3D (Xu et al., 2017) | 12.7 |
| Sigurdsson et al. (Sigurdsson et al., 2016a) | 12.8 |
| I3D baseline | 17.2 |
| I3D + 3 temporal conv. layers ($L = 5$) | 17.5 |
| I3D + 3 temporal conv. layers ($L = 30$) | 12.5 |
| I3D + LSTM | 18.1 |
| I3D + fixed temporal pyramid | 18.2 |
| I3D + super-events (Piergiovanni & Ryoo, 2018b) | 19.4 |
| I3D + 3 TGMs ($L = 5$) | 20.6 |
| I3D + 3 TGMs ($L = 30$) | 21.5 |
| I3D + 3 TGMs ($L = 5$) + super-events | 21.8 |
| I3D + 3 TGMs ($L = 30$) + super-events | **22.3** |

Joao Carreira and Andrew Zisserman. Quo vadis, action recognition? a new model and the kinetics dataset. In *Proceedings of the IEEE Conference on Computer Vision and Pattern Recognition (CVPR)*, 2017.

Achal Dave, Olga Russakovsky, and Deva Ramanan. Predictive-corrective networks for action detection. *arXiv preprint arXiv:1704.03615*, 2017.

Jeffrey Donahue, Lisa Anne Hendricks, Sergio Guadarrama, Marcus Rohrbach, Subhashini Venugopalan, Kate Saenko, and Trevor Darrell. Long-term recurrent convolutional networks for visual recognition and description. In *Proceedings of the IEEE Conference on Computer Vision and Pattern Recognition (CVPR)*, pp. 2625–2634, 2015.

Victor Escorcia, Fabian Caba Heilbron, Juan Carlos Niebles, and Bernard Ghanem. Daps: Deep action proposals for action understanding. In *Proceedings of European Conference on Computer Vision (ECCV)*, pp. 768–784. Springer, 2016.

Christoph Feichtenhofer, Axel Pinz, and Andrew Zisserman. Convolutional two-stream network fusion for video action recognition. In *Proceedings of the IEEE Conference on Computer Vision and Pattern Recognition (CVPR)*, pp. 1933–1941, 2016.

Chunhui Gu, Chen Sun, Sudheendra Vijayanarasimhan, Caroline Pantofaru, David A. Ross, George Toderici, Yeqing Li, Susanna Ricco, Rahul Sukthankar, Cordelia Schmid, and Jitendra Malik. AVA: A video dataset of spatio-temporally localized atomic visual actions. *arXiv preprint arXiv:1705.08421*, 2017.

Kensho Hara, Hirokatsu Kataoka, and Yutaka Satoh. Learning spatio-temporal features with 3d residual networks for action recognition. In *Proceedings of the ICCV Workshop on Action, Gesture, and Emotion Recognition*, volume 2, pp. 4, 2017.

Fabian Caba Heilbron, Victor Escorcia, Bernard Ghanem, and Juan Carlos Niebles. Activitynet: A large-scale video benchmark for human activity understanding. In *Proceedings of the IEEE Conference on Computer Vision and Pattern Recognition (CVPR)*, pp. 961–970, 2015.

Y.-G. Jiang, J. Liu, A. Roshan Zamir, G. Toderici, I. Laptev, M. Shah, and R. Sukthankar. THUMOS challenge: Action recognition with a large number of classes. http://crcv.ucf.edu/THUMOS14/, 2014.

Andrej Karpathy, George Toderici, Sanketh Shetty, Thomas Leung, Rahul Sukthankar, and Li Fei-Fei. Large-scale video classification with convolutional neural networks. In *Proceedings of the IEEE Conference on Computer Vision and Pattern Recognition (CVPR)*, pp. 1725–1732, 2014.

Will Kay, Joao Carreira, Karen Simonyan, Brian Zhang, Chloe Hillier, Sudheendra Vijaya-narasimhan, Fabio Viola, Tim Green, Trevor Back, Paul Natsev, et al. The kinetics human action video dataset. *arXiv preprint arXiv:1705.06950*, 2017.

Diederik Kingma and Jimmy Ba. Adam: A method for stochastic optimization. *arXiv preprint arXiv:1412.6980*, 2014.

Joe Yue-Hei Ng, Matthew Hausknecht, Sudheendra Vijayanarasimhan, Oriol Vinyals, Rajat Monga, and George Toderici. Beyond short snippets: Deep networks for video classification. In *Proceedings of the IEEE Conference on Computer Vision and Pattern Recognition (CVPR)*, pp. 4694–4702. IEEE, 2015.

AJ Piergiovanni and Michael S. Ryoo. Fine-grained activity recognition in baseball videos. In *CVPR Workshop on Computer Vision in Sports*, 2018a.

AJ Piergiovanni and Michael S. Ryoo. Learning latent super-events to detect multiple activities in videos. In *Proceedings of the IEEE Conference on Computer Vision and Pattern Recognition (CVPR)*, 2018b.

AJ Piergiovanni, Chenyou Fan, and Michael S Ryoo. Learning latent sub-events in activity videos using temporal attention filters. In *Proceedings of the American Association for Artificial Intelligence (AAAI)*, 2017.

Michael S Ryoo, Brandon Rothrock, and Larry Matthies. Pooled motion features for first-person videos. In *Proceedings of the IEEE Conference on Computer Vision and Pattern Recognition (CVPR)*, pp. 896–904, 2015.

Zheng Shou, Dongang Wang, and Shih-Fu Chang. Temporal action localization in untrimmed videos via multi-stage cnns. In *Proceedings of the IEEE Conference on Computer Vision and Pattern Recognition (CVPR)*, pp. 1049–1058, 2016.

Zheng Shou, Jonathan Chan, Alireza Zareian, Kazuyuki Miyazawa, and Shih-Fu Chang. Cdc: Convolutional-de-convolutional networks for precise temporal action localization in untrimmed videos. *arXiv preprint arXiv:1703.01515*, 2017.

Gunnar A Sigurdsson, Santosh Divvala, Ali Farhadi, and Abhinav Gupta. Asynchronous temporal fields for action recognition. *arXiv preprint arXiv:1612.06371*, 2016a.

Gunnar A. Sigurdsson, Gül Varol, Xiaolong Wang, Ali Farhadi, Ivan Laptev, and Abhinav Gupta. Hollywood in homes: Crowdsourcing data collection for activity understanding. In *Proceedings of European Conference on Computer Vision (ECCV)*, 2016b.

Karen Simonyan and Andrew Zisserman. Two-stream convolutional networks for action recognition in videos. In *Advances in Neural Information Processing Systems (NIPS)*, pp. 568–576, 2014.

Christian Szegedy, Vincent Vanhoucke, Sergey Ioffe, Jon Shlens, and Zbigniew Wojna. Rethinking the inception architecture for computer vision. In *Proceedings of the IEEE Conference on Computer Vision and Pattern Recognition (CVPR)*, pp. 2818–2826, 2016.

Du Tran, Lubomir D Bourdev, Rob Fergus, Lorenzo Torresani, and Manohar Paluri. C3d: generic features for video analysis. *CoRR, abs/1412.0767*, 2(7):8, 2014.

Du Tran, Jamie Ray, Zheng Shou, Shih-Fu Chang, and Manohar Paluri. Convnet architecture search for spatiotemporal feature learning. *arXiv preprint arXiv:1708.05038*, 2017.

Du Tran, Heng Wang, Lorenzo Torresani, Jamie Ray, Yann LeCun, and Manohar Paluri. A closer look at spatiotemporal convolutions for action recognition. In *Proceedings of the IEEE Conference on Computer Vision and Pattern Recognition (CVPR)*, pp. 6450–6459, 2018.

Ehsan Variani, Erik McDermott, and Georg Heigold. A gaussian mixture model layer jointly optimized with discriminative features within a deep neural network architecture. In *Acoustics, Speech and Signal Processing (ICASSP), 2015 IEEE International Conference on*, pp. 4270–4274. IEEE, 2015.

Gül Varol, Ivan Laptev, and Cordelia Schmid. Long-term Temporal Convolutions for Action Recognition. *IEEE Transactions on Pattern Analysis and Machine Intelligence*, 2017.

Huijuan Xu, Abir Das, and Kate Saenko. R-c3d: Region convolutional 3d network for temporal activity detection. *arXiv preprint arXiv:1703.07814*, 2017.

Serena Yeung, Olga Russakovsky, Ning Jin, Mykhaylo Andriluka, Greg Mori, and Li Fei-Fei. Every moment counts: Dense detailed labeling of actions in complex videos. *International Journal of Computer Vision (IJCV)*, pp. 1–15, 2015.

Serena Yeung, Olga Russakovsky, Greg Mori, and Li Fei-Fei. End-to-end learning of action detection from frame glimpses in videos. In *Proceedings of the IEEE Conference on Computer Vision and Pattern Recognition (CVPR)*, pp. 2678–2687, 2016.

Christopher Zach, Thomas Pock, and Horst Bischof. A duality based approach for realtime tv-l 1 optical flow. In *Joint Pattern Recognition Symposium*, pp. 214–223. Springer, 2007.

Yue Zhao, Yuanjun Xiong, Limin Wang, Zhirong Wu, Xiaoou Tang, and Dahua Lin. Temporal action detection with structured segment networks. *arXiv preprint arXiv:1704.06228*, 2017.

## A    IMPLEMENTATION DETAILS

As our base per-segment CNN, we use the I3D (Carreira & Zisserman, 2017) network pretrained on the ImageNet and Kinetics (Kay et al., 2017) datasets. I3D obtained state-of-the-art results on segmented video tasks, and this allows us to obtain reliable $v_t$. We also use two-stream version of InceptionV3 (Szegedy et al., 2016) pretrained on Imagenet and Kinetics as our base per-frame CNN, and compared them. We chose InceptionV3 as it is deeper than previous two-stream CNNs such as (Simonyan & Zisserman, 2014; Feichtenhofer et al., 2016). We extracted frames from the videos at 25 fps, computed TVL1 (Zach et al., 2007) optical flow, clipped to $[-20, 20]$. For InceptionV3, we computed features for every 3 frames (8 fps). For I3D, every frame was used as the input. I3D has a temporal stride of 8, resulting in 3 features per second (3 fps).

We implemented our TGM layers as well as other baseline layers in PyTorch. Our default setting was as follows: for 3-layer models, we set $L = 10$ for frame-based features (i.e., InceptionV3) and $L = 5$ for segment-based features (i.e., I3D), as each segment already contains some temporal information. For 1-layer models, we set $L = 30$ for frame-based features and $L = 15$ for segment-based features. We set $M = 16$ and $C_{out} = 80$ and $C_{out} = 65$ for the last TGM layer. We found these values to work well on a held out portion of the training set of MultiTHUMOS. In all models, we used one fully-connected layer at the end to make the per-frame or per-segment classification.

We trained our models using the Adam (Kingma & Ba, 2014) optimizer with the learning rate set to 0.01. We decayed the learning rate by a factor of 10 after every 10 training epochs. We trained our models for 50 epochs. We plan to make all our source code and trained models publicly available once the paper is published.

## B    HYPERPARAMETER EXPERIMENTS

We conducted a set of experiments to compare the effects of the temporal duration, $L$, number of Gaussians, $M$, and the number of output channels, $C_{out}$. For these experiments, we only used the one-stream version of I3D with RGB inputs.

**Effect of $L$:**    In Table 6, we compare different values of $L$. For these experiments, we use $M = 16$ and $C_{out} = 16$. We find that the 3-layer model with $L = 5$ performs the best. With I3D features, this allows the model to capture up to 8 seconds of information. The average activity in MultiTHUMOS is 3.3 seconds long and the maximum is 14.7 seconds long, and with this setting, the model is able to capture enough temporal context to perform well. Larger values of $L$ capture too much temporal information, but due to the Gaussian structure, it does not drastically harm performance. Figure 6 shows that even with longer kernels, the Gaussians learn to focus mostly on the center of the interval and capture the rough duration of the activities. Thus, having too long intervals does not drastically harm performance, which is in contrast to the standard 1-D convolution. Note that for Charades, the temporal kernels are learned to capture much longer temporal duration, as the average activity in charades is 12.8 seconds and larger values of $L$ perform better.

Figure 6 illustrates examples of the learned TGM kernels of various lengths. The figure shows that the kernels focus on short temporal intervals on MultiTHUMOS even if we make the filters longer, as the activities are an average of 3.3 seconds long. On Charades, the TGM kernels learn to capture much longer intervals, as the activities are an average of 12.8 seconds long. We believe that this suggests TGMs are learning to capture information from the important necessary intervals.

In Table 6, we also report the results of using a standard 1-D conv. layer with different $L$ values. The number of parameters in our TGM layer is independent of $L$, however, with the standard 1-D conv. layer, the number of parameters increases as $L$ increases. We find that increasing $L$ with 1-D convolution helps for small values of $L$, but for $L > 15$, the performance drastically drops, while TGM layers only show a small decrease.

**Effect of $M$:**    In Table 7, we compare different values of $M$. For these experiments, we set $L = 15$ and $C_{out} = 16$. We find that $M = 16$ performs best, suggesting that smaller values of $M$ restrict the possible temporal kernels too much. We also observe that larger values of $M$ performs slightly worse than $M = 16$ (but not much), likely because they introduce more parameters than needed. When $M$ and $L$ have similar values, it allows the model to learn a sufficient number of Gaussians

Table 6: Effect of $L$ on MultiTHUMOS and Charades using only RGB I3D features. Note that the 3 TGM layer models have larger temporal resolution than the 1 TGM layer models for the same values of $L$. We also compare to using standard one-layer 1-D conv layer with different values of $L$.

| | MultiTHUMOS | | | Charades | | |
|---|---|---|---|---|---|---|
| | 1 Layer | 3 Layers | 1-D Conv | 1 Layer | 3 Layers | 1-D Conv |
| I3D Baseline | 22.3 | - | - | 15.3 | - | - |
| $L = 3$ | 30.2 | 31.7 | 26.6 | 15.5 | 16.1 | 15.5 |
| $L = 5$ | 32.5 | **37.2** | 28.3 | 15.7 | 17.8 | 16.3 |
| $L = 10$ | 34.5 | 35.4 | 31.7 | 16.1 | 18.2 | 16.6 |
| $L = 15$ | **36.1** | 34.1 | 32.5 | 17.5 | 18.6 | 16.8 |
| $L = 30$ | 32.5 | 33.9 | 26.5 | 18.1 | **18.9** | 12.1 |
| $L = 50$ | 32.1 | 33.7 | 15.4 | **18.3** | 18.8 | 6.7 |

Table 7: Comparison of various values of $M$ on MultiTHUMOS and Charades using RGB I3D features. For these experiments, 1 layer was used with $L = 15$ and $C_{out} = 16$.

| | MultiTHUMOS | Charades |
|---|---|---|
| $M = 2$ | 27.8 | 15.5 |
| $M = 4$ | 33.1 | 16.2 |
| $M = 8$ | 34.8 | 17.5 |
| $M = 16$ | 36.1 | 17.5 |
| $M = 32$ | 35.7 | 17.1 |
| $M = 64$ | 35.8 | 17.3 |

Table 8: Comparison of values of $C_{out}$ on MultiTHUMOS and Charades using RGB I3D features. For these experiments, 1 layer was used with $L = 15$ and $M = 16$.

| | MultiTHUMOS | Charades |
|---|---|---|
| $C_{out} = 1$ | 33.5 | 16.2 |
| $C_{out} = 4$ | 34.2 | 17.4 |
| $C_{out} = 8$ | 35.5 | 17.5 |
| $C_{out} = 16$ | 36.1 | 17.5 |
| $C_{out} = 32$ | 36.0 | 17.2 |
| $C_{out} = 64$ | 36.1 | 17.4 |
| $C_{out} = 80$ | 36.1 | 17.5 |

and create a diverse range of temporal kernels. When $M$ is larger than $L$, it results in learning a kernel similar to standard 1-D convolution.

**Effect of $C_{out}$:** In Table 8, we compare different values of $C_{out}$. For these experiments, $L = 15$, we used 1-layer and $M = 16$. We find that $C_{out}$ performs best when set to 16 or larger on these datasets. Larger values of $C_{out}$ seem to capture redundant information, as it does not lower performance.

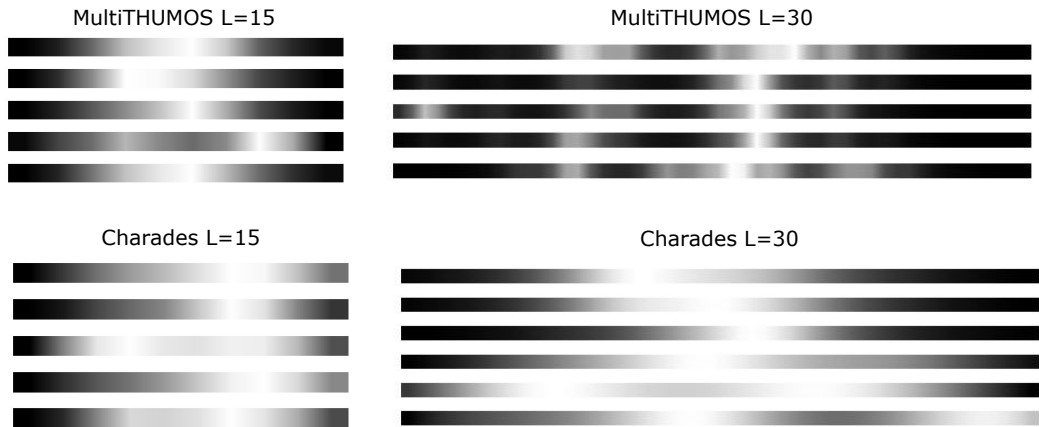

Figure 6: Illustration of several learned TGM kernels. On MultiTHUMOS, it learns to focus on shorter intervals to capture shorter events. On Charades, the Gaussians have a larger $\sigma$ value, resulting in filters that attend to longer temporal durations.

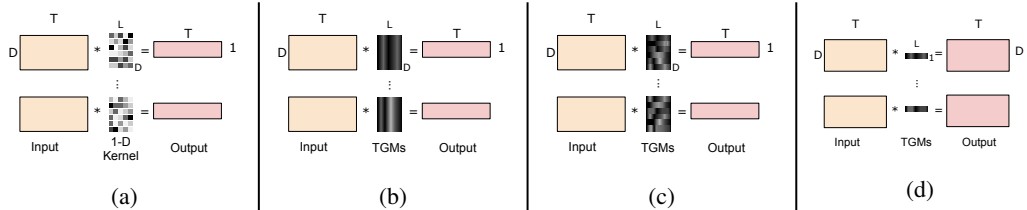

(a)      (b)      (c)      (d)

Figure 7: **(a-c)** Different forms of 1-D temporal convolutions which take a $D \times T$ input and produces a $C \times T$ output based on $C$ number of $D \times L$ kernels: **(a)** the standard 1-D convolution, **(b)** using Gaussian mixtures for 1-D convolution while sharing Gaussian mixtures across input channels, and **(c)** using $D$ different Gaussian mixtures for 1-D convolution. **(d)** Our TGM layer in its simplest form (i.e., 1-layer case) applying the $1 \times L$ temporal kernel in a 2-D convolutional fashion, maintaining both time and feature axis.

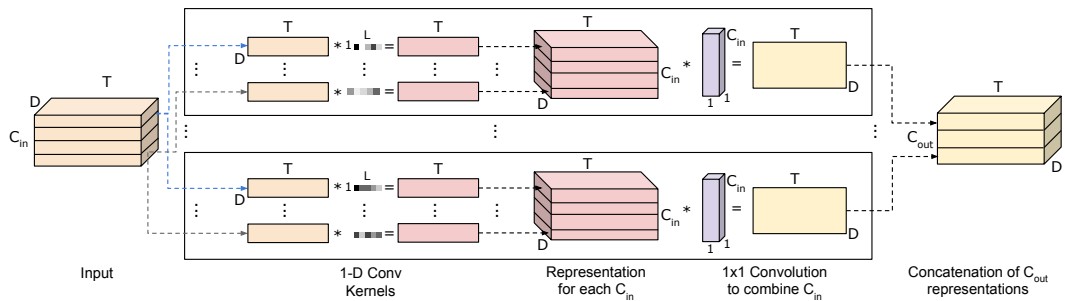

Figure 8: A temporal convolutional layer with channel combination similar to Fig. 3. The difference is that this layer does not learn Gaussian mixtures, but unconstrained 1-D temporal kernels.

## C    COMPARISON OF DIFFERENT LAYER FORMS

To confirm the various aspects of our design, we conducted experiments comparing different types of temporal convolution. In Fig. 7a we illustrate the standard 1-D convolution, taking $D \times T$ input and producing a $C \times T$ output, where $D$ is the number of input channels and $C$ is the number of output channels. In Fig. 7b, we illustrate the method of applying a Gaussian mixture kernel as 1-D convolution. Here, the Gaussian mixture kernel is shared by all $D$ input channels and we learn a $C$ number of such kernels. In Fig. 7c, we illustrate the approach of applying a Gaussian mixture kernel as 1-D convolution while learning $D$ different Gaussian mixtures. This is very similar to the standard 1-D convolution, except that the filter values are constrained to have the shape of Gaussian mixtures.

Fig. 8 illustrates one more baseline. This is similar to our full TGM layer with the channel-combination described Fig. 3. However, in this baseline, instead of learning Gaussian mixtures, we learn $C_{in} \cdot C_{out}$ number of $1 \times L$ kernels. The kernel values are left unconstrained. While the TGM layer has $2 \cdot M + C_{in} \cdot C_{out} \cdot M + C_{in} \cdot C_{out}$ parameters, this layer has $L \cdot C_{in} \cdot C_{out} \cdot M + C_{in} \cdot C_{out}$, which is more than the TGM layer.

In Table 9, we compare the results of the various above-mentioned layers on MultiTHUMOS using RGB I3D features. We find that the Fig. 7b method performs poorly, while the Fig. 7c method slightly outperforms the standard 1-D convolution. The Fig. 8 method is slightly better than the standard 1-D convolution, but performs worse than Fig. 7c. However, none of these layers perform as well as our TGM layer, confirming that both the design of learning Gaussian mixtures and maintaining temporal channel axis are important for activity detection.

Table 9: Comparison of the different forms of temporal convolution on MultiTHUMOS using RGB I3D features. We set $L = 15$ and used 1 layer models for these experiments.

|  | MultiTHUMOS |
| --- | --- |
| Standard 1-D Convolution (Fig. 7a) | 32.5 |
| The layer described in Fig. 7b | 28.6 |
| The layer described in Fig. 7c | 33.2 |
| The layer described in Fig. 8 | 32.8 |
| Our TGM Layer | 36.1 |

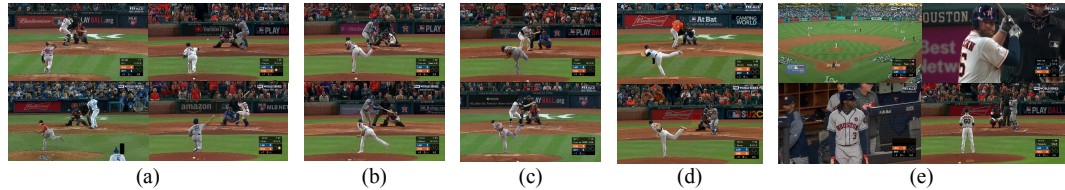

|       |       |       |       |       |
| ----- | ----- | ----- | ----- | ----- |
| (a)   | (b)   | (c)   | (d)   | (e)   |

Figure 9: Examples of several of the activities in the MLB-YouTube dataset: (a) Pitch, (b) Hit, (c) Bunt, (d) Hit by pitch, (e) No activity. This shows the difficulty of this dataset, as the difference between hit and bunt, swing and no swing are very small.

## D    EXPERIMENTS ON ADDITIONAL DATASETS

### D.1    MLB-YOUTUBE DATASET

#### D.1.1    DATASET

The MLB-YouTube dataset (Piergiovanni & Ryoo, 2018a) consists of 20 baseball games from the 2017 MLB post-season available on YouTube. This dataset consists of over 42 hours of video. For these experiments, we used the continuous video setting which have 2,126 1-2 minute long clips. Each clip is densely annotated with the baseball activities that occur. There are 8 activity classes: pitch, strike, ball, swing, hit, foul, hit by pitch, and bunt. Examples of some of these classes are shown in Fig. 9. Each continuous clip contains on average of 7.2 activities, giving a total of over 15,000 activity instances in the dataset.

What makes this dataset challenging is that the variation between classes is very small. In ActivityNet (Heilbron et al., 2015), for example, the difference between swimming and brushing hair is drastic. The background, motion, and even size of the person in the video is different. However, in broadcast baseball videos, the difference between a ball and a strike, or a swing and a bunt, are small. All actions are recorded from the same camera angle as we can confirm from Fig. 9.

#### D.1.2    RESULTS

In Table 10, we compare various approaches on this dataset. Our TGM layers improve over the baseline by ~6% (40.1 vs. 34.2). Additionally, we compare to methods using the super-event representation (Piergiovanni & Ryoo, 2018b), which previously achieved state-of-the-art performance on several activity detection datasets. On this dataset, our approach outperforms the super-event representation, and further the concatenation of our TGM representation with such super-event representation performs best by a significant margin (~13% compared to the baseline). This suggests that TGMs and super-event capture different temporal information and are both useful to the detection task.

We further find that using multiple, standard temporal convolution layers leads to worse performance, likely due to overfitting from the large number of parameters. While using multiple TGM layers improves performance, confirming that the Gaussian structure and sparsity constraint benefits model learning.

Table 10: Result mAP on the MLB-YouTube dataset using InceptionV3 and I3D to obtain features. Our TGM layers significantly outperform the baseline models.

| Model | Spatial | Temporal | Two-stream |
|---|---|---|---|
| Random | 13.4 | 13.4 | 13.4 |
| InceptionV3 | 31.2 | 31.8 | 31.9 |
| InceptionV3 + LSTM | 32.1 | 33.5 | 34.1 |
| InceptionV3 + 1 temporal conv | 32.8 | 34.4 | 35.2 |
| InceptionV3 + 3 temporal conv | 28.4 | 29.8 | 30.1 |
| InceptionV3 + super-events | 31.5 | 36.2 | 39.6 |
| InceptionV3 + 1 TGM | 32.4 | 36.3 | 37.4 |
| InceptionV3 + 3 TGM | 33.2 | 38.2 | 38.2 |
| InceptionV3 + 3 TGM+super-events | 34.6 | 42.4 | 42.9 |
| I3D | 33.8 | 35.1 | 34.2 |
| I3D + LSTM | 36.2 | 37.3 | 39.4 |
| I3D + 1 temporal conv | 37.3 | 38.6 | 39.9 |
| I3D + 3 temporal conv | 32.4 | 34.6 | 35.6 |
| I3D + super-events | 38.7 | 38.6 | 39.1 |
| I3D + 1 TGM | 35.5 | 37.5 | 38.5 |
| I3D + 3 TGM | 36.5 | 38.4 | 40.1 |
| I3D + 3 TGM+super-events | 39.4 | 46.0 | **47.1** |

Table 11: Results on AVA dataset with the temporal annotation-only setting (i.e., frame classification without using bounding box training labels).

| | mAP |
|---|---|
| Random | 2.65 |
| I3D baseline | 7.5 |
| I3D + 3 temporal conv. layers | 7.9 |
| I3D + LSTM | 7.8 |
| I3D + super-events(Piergiovanni & Ryoo, 2018b) | 9.8 |
| I3D + 1 TGMs | 11.2 |
| I3D + 3 TGMs | 14.5 |
| I3D + 3 TGMs + super-events | 14.9 |

## D.2 AVA

### D.2.1 DATASET

AVA (Gu et al., 2017) is a large-scale video dataset containing of 80 atomic action classes in 57k video clips. These clips are drawn from movies. Existing datasets, such as Charades, have very specific actions that depend on objects, such as holding a cup vs. holding a picture. In AVA, the actions are intentionally generic, such as sit, stand, hold, carry, etc. Further, the AVA dataset is annotated with both spatial and temporal locations of activities. Since we are interested in temporal activity detection, we follow the setting of Piergiovanni & Ryoo (2018b) and label each frame with the occurring activities while ignoring the spatial location. We evaluate performance following the same method as MultiTHUMOS, Charades and MLB-YouTube by measuring per-frame mAP.

### D.2.2 RESULTS

In Table 11, we present the results of our model. We again find that temporal convolution and LSTMs provide some benefit over the baseline, but TGM layers further improve performance. Again, combining the TGM, which captures local temporal structure, with super-events which capture global temporal structure, provides the best performance by $\sim 7.4\%$.

