# OpenReview forum: "Temporal Gaussian Mixture Layer for Videos"
_ICLR.cc/2019/Conference_

### Official Review · AnonReviewer3 · 2018-10-30
**Good paper, need more justification on the learned features by TGM**

**Rating:** 7
**Confidence:** 5

**Review:**

This paper introduces a new convolutional layer named the Temporal Gaussian Mixture (TGM) layer, and present how it can be used for activity recognition. The kernels of the new layer are controlled by a set of (temporal) Gaussian distribution parameters, which significantly reduce learnable parameters. The results are complete on four benchmarks and show consistent improvement. I just have minor comments.

1. I am curious what the learned feature look like. As the author mentioned, "The motivation is to make each temporal Gaussian distribution specify (temporally) ‘where to look’ with respect to the activity center, and represent the activity as a collection/mixture of such temporal Gaussians convolved with video features." So does the paper achieve this goal?

Another thing is, can the authors extract the features after TGM layer, and maybe perform action recognition on UCF101 to see if the feature really works? I just want to see some results or visualizations to have an idea of what TGM is learning.

2. How long can the method actually handle? like hundreds of frames? Since the goal of the paper is to capture long term temporal information.

3. It would be interesting to see an application to streaming video. For example, surveillance monitoring of human activities.

---

> ### Author Response · Authors · 2018-11-10
> **Added visualizations to the appendix**
>
> Thank you for your review. We have updated the paper to address your questions.
>
> 1) Filter visualization:
>
> We have added a figure to the appendix (Figure 6) illustrating examples of the learned TGM kernels. The most interesting aspect of this figure is that it shows the kernels focus on short temporal intervals on MultiTHUMOS even if we make the filters long, as the activities are an average of 3.3 seconds long. On Charades, the TGM kernels learn to capture much longer intervals, as the activities are an average of 12.8 seconds long. We believe that this suggests TGMs are learning to capture information from the important necessary intervals.
>
> TGMs aren't really suitable for datasets like UCF101, as those videos are quite short (2-3 seconds on average). CNNs like I3D already capture 2-3 seconds of temporal information, so there isn't much benefit to applying TGMs there.
>
> 2) Our approach is able to handle continuous (streaming) videos of any duration by its design. Average lengths of videos in MultiTHUMOS is around 2.5 minutes and those in Charades is around 30 seconds. Within these videos, the TGM layers are able to capture the longer temporal structure than prior works. Using L=30 in the 3 layer model on Charades with I3D, we are able to capture ~800 frames (roughly +-15 seconds from each frame) of temporal information. In contrast, I3D only captures 99 frames (+- 2 seconds).
>
> 3) As our TGM layer is (temporally) convolutional, it can easily be applied to streaming video in an online, convolutional fashion.
>
> Please let us know if this answers your questions and if you have any others.

---

### Official Review · AnonReviewer2 · 2018-10-31
**TGMs are an interesting idea but need better justification and comparisons to standard 1D convolutions**

**Rating:** 6
**Confidence:** 5

**Review:**

Summary:
This paper proposes a temporal convolution layer called Temporal Gaussian Mixture (TGM) for video classification tasks. Usually, video classification models can be decomposed into a feature extraction step, in which frames are processed separately or in local groups, and a classification step where labels are assigned considering the full video features/longer range dependencies. Typical choices for the classification step are Recurrent Neural Networks (RNNs) and variations, pooling layers and 1D temporal convolutions. TGMs are 1D temporal convolutions with weights defined by taking samples from a mixture of Gaussians. This allows to define large convolutional kernels with a reduced number of parameters -the mean and std of the gaussians- at the cost of having reduced expressiveness. The authors experiment replacing 1D temporal convolutions with TGMs in standard video classifications models and datasets and report state-of-the-art results.

Strengths:
[+] The idea of defining weights as samples from a Mixture of Gaussians is interesting. It allows to define convolutional kernels that scale with the number of channels (equivalent to the number of Gaussians) instead of the number of channels and receptive field.

Weaknesses:
[-] The explanation of the TGM layers is very confusing in its current state.

Certain aspects of TGMs that would help understand them are not clearly stated in the text. For example, it is not clear that 1) TGMs are actually restricted 1D convolutions as the kernel is expressed through a mixture of gaussians, 2) In a TGM the number of mixtures corresponds to the number of output channels in a standard 1D convolution and 3) despite TGMs are 1D convolutions, they are applied differently by keeping the number of frame features (output channels of a CNN) constant through them. Section 3 does not explain any of these points clearly and induces confusion.

[-] The comparisons in the experimental section are unfair for the baselines and do not justify the advantages of TGMs.

TGMs are 1D convolutions but they are applied to a different axis - 1D convolutions take as input channels the frame features. Instead, TGMs add a dummy axis, keep the number of frame features constant through them, and finally use a 1x1 convolution to map the resulting tensor to the desired number of classes. There is no reason why TGMs can’t be used as standard 1D convolutions taking as the first input channels the frame features - in other words, consider as many input mixture components as frame features for the first TGM. This is a crucial difference because the number of parameters of TGMs and the baselines is greatly affected by it.

The implicit argument in the paper is that TGMs perform better than standard 1D convolutions because they have less parameters. However, in the experiments they compare to 1D convolutions that have smaller temporal ranges – TGMs have a temporal kernel of 30 timesteps or 10 per layer (in a stack of 3 TGMs), whereas the 1D convolutions either have 5 or 10 timestep kernels according to section 4. Thus, it is impossible to know whether TGMs work better because 1) in these experiments they have longer temporal range – we would need a comparison to 1D convolutions applied in the same way and with the same temporal range to clarify it, 2) because they are applied differently – we would need a comparison when TGMs are used the same way as 1D convolutions, 3) because the reduced number of parameters leads to less overfitting – we would need to see training metrics and see the generalization gap when compared to equivalent 1D convolutions or 4) because the reduced number of parameters eases the optimization process. To sum up, it's true that TGMs would have a reduced number of parameters compared to the equivalent 1D convolutions with the same temporal range, but how does that translate to better results and, in this case, why is the comparison made with non-equivalent 1D convolutions? What would happen if the authors compared to 1D convolutions used in the same way as TGMs?

[-] To claim SOTA, there are comparisons missing to recently published methods.
In particular, [1] and [2] are well-known models that report metrics for the Charades dataset also used in this paper.

Rating:
At this moment I believe the TGM layer, while a good idea with potential, is not sufficiently well explained in the paper and is not properly compared to other baselines. Thus, I encourage the authors to perform the following changes:

-Rewrite section 3, better comparing to 1D convolutions and justifying why TGMs are not used in the same manner but instead using a dummy axis and a linear/1x1 conv layer at the end and what are the repercussions of it.
-Show a fair comparison to 1D convolutions as explained above.
-If claiming SOTA results, compare to [1] or [2] which use different methods to the one proposed in these paper but outperform the baselines used in the experiments.

[1] Wang, Xiaolong, et al. "Non-local neural networks." CVPR 2018
[2] Zhou, Bolei et al. "Temporal Relational Reasoning in Videos" ECCV 2018


----------------------
Update: In light of the authors' rebuttal I have updated my rating from 5 to 6.

---

> ### Author Response · Authors · 2018-11-06
> **Regarding the comment on the experimental comparison being unfair due to TGM kernels using larger temporal size**
>
> We thank the reviewer for the detailed comments. We will get back to you later with more detailed answers/revisions to address your concerns. Here, we would like to clarify one important confusion/misunderstanding as soon as possible.
>
> - Regarding the comment on the experimental comparison being unfair due to TGM kernels using larger temporal size:
>
> In all our experiments, the TGM layers, standard 1D convolutional layers, and temporal pyramid pooling layers used the filters with the identical temporal length (i.e., L). To be more specific, when using InceptionV3 as the base architecture (obtained at 8fps), our L was 30 for 1-layer models and 10 for 3-layer models. When using I3D as the base architecture (obtained at 3fps), our L was 15 for 1-layer models and 5 for 3-layer models. This was the case for both the TGM layers and the standard 1D conv layers.
>
> We will revise the paper to clarify this more explicitly. The current version of the paper describes that L=30 or 10 in Section 4.1 (we meant for InceptionV3) while also indicating that L=10 for InceptionV3 and L=5 for I3D in Appendix A, which we agree is confusing.

---

> ### Author Response · Authors · 2018-11-10
> **Revised section 3 and added more comparisons in the appendix**
>
> Thank you for your detailed comments. We have updated the paper to address your concerns.
>
> 1) Explanation of the TGM layers:
>
> We have revised Section 3 to clarify our approach and better compare to 1-D convolution and justify our design decisions. In addition, we added a new figure in the appendix (Figure 7), to help the understanding of the difference between the TGM layer, the standard 1D convolution, and other forms of using Gaussian mixtures. Please let us know if you find the revised version helpful.
>
> 2) Comparison to more baselines and clarification of our L setting:
>
> As we mentioned in the previous comment, we revised the paper to clarify our L setting. All the baselines and the TGM layers use the identical default L setting of L=30/10 (for InceptionV3) and L=15/5 (for I3D), for 1-layer/3-layers, unless otherwise mentioned. We revised the paper to make this explicit.
>
> In Appendix C we added more experiments comparing different forms of the Gaussian mixture layers as suggested (Table 9). These include learning of temporal Gaussian mixtures to be convolved in the standard 1-D convolution fashion without temporal channel axis (Figure 7b-c), and a temporal convolutional layer with channel combination but without Gaussians (Figure 8). We found that none of these alternatives performed as well as our TGM layer, confirming that both the Gaussian mixture and added temporal channel axis are important for the performance.
>
> 3) Comparison to Non-local neural networks and Temporal Relational Reasoning networks:
>
> We believe there is a misunderstanding/confusion and we want to clarify it.
>
> We want to emphasize that we are not claiming state-of-the-art accuracy on activity 'classification', but claiming the state-of-the-art  accuracy on temporal activity 'detection' from continuous videos. Classification and detection (often also called 'localization') are relevant but different problems. Both non-local neural networks and Temporal Relational Reasoning focuses on the activity classification problem and they only report their performances on the classification datasets. To our knowledge, there is no prior work reporting better detection accuracy than ours on the continuous video public datasets we used: MultiTHUMOS and the Charades_v1_localize setting of Charades.
>
> To be more specific, non-local neural networks and Temporal Relational Reasoning networks both test on the 'classification' setting of Charades, while we are focusing on the 'localization' setting (i.e., dense labeling). Note that Charades has two different evaluation settings and maintain the results of these two settings separately.
>
> One may argue to extend such classification works for the detection, but there are limitations. Due to the design of the non-local neural network, it is non-trivial to extend it to variable length, continuous videos as the non-local layer requires reshaping to a fixed THW tensor. This does not support variable length videos, which is necessary for dense-labeling detection tasks with variable length, continuous videos. Temporal Relation Reasoning networks were only evaluated on the classification setting and provided lower performance than the I3D we use as a baseline (25.2 for TRN vs. 34.4 for I3D). If it were extended to the localization setting, we believe it would become a weaker baseline than I3D.
>
> Further, notice that we can always apply TGM layers on top of any baseline CNNs (e.g., I3D or anything better) to increase the performance by capturing longer temporal information.
>
> Please let us know if this answers your questions and if you have any others.

---

> > ### Comment · AnonReviewer2 · 2018-12-04
> > **Answer to the author's comments**
> >
> > 1) Explanation of the TGM layers and 2) Comparison to more baselines and clarification of our L setting
> >
> > I still believe the description of TGM layers and how they are applied is a bit misleading. Essentially, TGMs are learning a 1-D convolutional kernel. The fact that they it was chosen to apply them in a different way (by using 3-D tensors and not modifying one of the dimensions) is orthogonal to the core operation they are performing.
> >
> > I believe this is clarification is quite important as, following the ablation study done comparing to 1-D convolutions applied in different ways, it seems that the way TGMs are applied is an important part to obtain better results. In its current form the paper justifies the success of TGMs on the reduced number of parameters compared to a standard 1-D convolution, but there is no discussion or justification for the impact of how TGM layers are applied.  Therefore, I think the ablation study should be part of main body of the paper (and not be found in the appendix) and should be more thoroughly discussed in a future version of the manuscript.
> >
> > As for the choice of L, given the new description I think that part of the comparison is fair. Note that in the initial version of the paper the description of the choice of L for different methods in the main paper was different than the current one and it was in conflict with the information in the appendix.
> >
> > 3) Comparison to Non-local neural networks and Temporal Relational Reasoning networks:
> > My comment was meant to compare TGMs to the cited papers on activity classification rather than adapt those methods to localization - seems like just changing the loss function of TGMs would allow that. I agree though the claims are about localization, and therefore the comparison is not fully justified and they claims are fair.

---

> > > ### Author Response · Authors · 2018-12-13
> > > **Revisions and Charades experiment**
> > >
> > >
> > > 1/2) We will further revise the paper to clarify the effect of having an additional temporal channel axis. We will move the ablation experiments we have in the appendix to the main paper, and add discussions on the impacts of different forms of convolutional layers.
> > >
> > > 3) Following the suggestion from the reviewer, we conducted an experiment using the TGMs for the classification task on Charade. This was done by temporally averaging the predictions made by the TGM model to obtain video-level predictions (similar to the I3D paper). Using 3 TGM layers, L=30 with the I3D features, we obtained mAP of 38.7 on the classification setting. This is better than TRN (25.2) and Non-local neural networks (37.5) using the same setting as well. We will include it in the paper.

---

### Official Review · AnonReviewer1 · 2018-11-01
**An Official Review: Temporal Gaussian Mixture Layer for Videos**

**Rating:** 6
**Confidence:** 3

**Review:**

This paper presents Temporal Gaussian Mixture (TGM) layer, efficiently capturing longer-term temporal dependencies with smaller number of parameters. The authors apply this layer to the activity recognition problem, claiming state-of-the-art performance compared against several baselines.

Strong points of this paper:
 - The authors clearly described the proposed layer and model step by step, first explaining TGM layer, followed by single layer model, then generalizing it to multi-layer model.
 - The authors achieved state-of-the-art performance on multiTHUMOS dataset. The results look great in two aspects: the highest MAP scores shown in Table 1, and significantly smaller number of parameters shown in Table 2 to achieve the MAP scores.

Questions:
 - Basically, the idea in this paper is proposing to parameterize conv layers with Gaussian mixtures, with multiple pairs of mean and variance. Although Gaussian mixtures are powerful to model many cases, it might not be always to perfectly model some datasets. If a dataset is highly multi-modal, Gaussian mixture model also needs to have large M (number of mixture components). It is not clear how the authors decided hyper-paramter M, so it will be nicer for authors to comment the effect of different M, on various dataset/task if possible.
 - Same for the hyper-parameter L, the temporal duration. It will be nicer to have some experiments with varied L, and to discuss how much this model is sensitive to the hyper-parameter.

---

> ### Author Response · Authors · 2018-11-10
> **Added experiments to the appendix**
>
> Thank you for your review and the suggestions. Please find our answers below to address your concerns. We also have updated the paper.
>
> 1) Effect of different M values on different datasets:
>
> We conducted a new set of experiments for this, and added them to the appendix. Appendix B shows experimental results comparing different values for M and C_out, which are the parameters that determine the number of Gaussians and mixtures. On both Charades and MultiTHUMOS, we find that M=16 and C_out>=16 performs best. Larger values of M lead to slightly lower performance, likely because they introduce more parameters and end up learning a kernel similar to a standard 1-D convolution. Smaller values of M limit the possible temporal kernels, which reduces performance quite a bit. Larger values of C_out do not impact performance, likely learning redundant information.
>
> 2) Effect of different L:
>
> Similarly, we added multiple experimental results comparing different values of L for models with 1 TGM layer, 3 TGM layers, and one 1-D Conv layer. We find that on MultiTHUMOS, setting L=15 (1 layer)/5 (3 layers) performs best (which was our default setting in the paper), as this captures about 8 seconds of information (i.e., +- 4 seconds) and activities are on average 3.3 seconds in MultiTHUMOS. However, on Charades, we found that larger values of L performed better. The average activity length is 12.8 seconds, and we found setting L=30 with 3 TGM layers performed best. This increased the overall performance of our {two-stream I3D + 3 TGM layers} model from 21.8 to 22.3. Since the number of parameters in our TGM layer is independent of L, we were able to increase the performance by using larger L values, while 1-D conv needs to learn many more parameters for larger L and it decreases performance. Thank you for suggesting these experiments.
>
> We have also added a figure in the appendix (Figure 6)  illustrating the learned TGM kernels which visually confirms that the TGM captures longer temporal information for Charades and shorter intervals for MultiTHUMOS.
>
> Please let us know if this answers your questions and if you have any others.

---

### Author Response · Authors · 2019-04-28
**The paper will be presented at ICML 2019.**

This paper, which got 3 accept reviews and got rejected from ICLR 2019, will appear at ICML 2019.

Thanks,

Michael

---

### Meta-Review · Area_Chair1 · 2018-12-20

**Confidence:** 5
**Recommendation:** Reject

**Metareview:**

The reviewers raised a number of major concerns including lack of explanations, lack of baseline comparisons, and lack of discussion on pros and cons of  the main contribution of this work --  the presented Temporal Gaussian Mixture (TGM) layer. The authors’ rebuttal addressed some of the reviewers’ comments but failed to address all concerns (especially when it comes to the success of TGMs; it remains unclear whether this could be attributed solely to the way TGMs are applied rather than to their fundamental methodological advantage). Having said that, I cannot suggest this paper for presentation at ICLR.